# Dynamic relational event modeling: Testing, exploring, and applying

**Marlyne Meijerink-Bosman**[1]*, **Roger Leenders**[2,3], **Joris Mulder**[1,3]

**1** Department of Methodology & Statistics, Tilburg University, Tilburg, The Netherlands, **2** Department of Organization Studies, Tilburg University, Tilburg, The Netherlands, **3** Jheronimus Academy of Data Science, 's-Hertogenbosch, The Netherlands

* m.l.meijerink@tilburguniversity.edu

**Data Availability Statement:** All data and script for the analyses are available from https://github.com/mlmeijerink/REHdynamics.

**Funding:** This research was supported by a VIDI grant from the Netherlands Organization for

## Abstract

The relational event model (REM) facilitates the study of network evolution in relational event history data, i.e., time-ordered sequences of social interactions. In real-life social networks it is likely that network effects, i.e., the parameters that quantify the relative importance of drivers of these social interaction sequences, change over time. In these networks, the basic REM is not appropriate to understand what drives network evolution. This research extends the REM framework with approaches for testing and exploring time-varying network effects. First, we develop a Bayesian approach to test whether network effects change during the study period. We conduct a simulation study that illustrates that the Bayesian test accurately quantifies the evidence between a basic ('static') REM or a dynamic REM. Secondly, in the case of the latter, time-varying network effects can be studied by means of a moving window that slides over the relational event history. A simulation study was conducted that illustrates that the accuracy and precision of the estimates depend on the window width: narrower windows result in greater accuracy at the cost of lower precision. Third, we develop a Bayesian approach for determining window widths using the empirical network data and conduct a simulation study that illustrates that estimation with empirically determined window widths achieves both good accuracy for time intervals with important changes and good precision for time intervals with hardly any changes in the effects. Finally, in an empirical application, we illustrate how the approaches in this research can be used to test for and explore time-varying network effects of face-to-face contacts at the workplace.

## Introduction

Relational event history data consist of time-ordered sequences of events between individuals [1]. For example, the relational event history in Table 1 consists of face-to-face contacts between employees in the workplace [2]. For each event in the relational event history we observe the time point and the individuals who are involved. Since relational event history data capture the timing and sequencing of social interactions on a fine-grained timescale, this type of data contain detailed information that helps us learn about interaction dynamics in

Scientific Research (NWO, https://www.nwo.nl/) number 452-17-006, received by Joris Mulder. The funders had no role in study design, data collection and analysis, decision to publish, or preparation of the manuscript.

**Competing interests:** The authors have declared that no competing interests exist.

**Table 1. The first 10 events from a relational event history with face-to-face contacts between employees in the workplace.**

| time | employee 1 | employee 2 |
|------|------------|------------|
| 08:00:40 | 0574 | 1362 |
| 08:00:40 | 0164 | 0779 |
| 08:01:00 | 0447 | 0763 |
| 08:01:00 | 0117 | 0429 |
| 08:01:40 | 0215 | 1414 |
| 08:01:40 | 0097 | 1204 |
| 08:01:40 | 0461 | 1245 |
| 08:01:40 | 0020 | 1209 |
| 08:01:40 | 0015 | 0020 |
| 08:02:00 | 0020 | 0985 |

social networks. The relational event model (REM) [1] analyzes relational event history data in a direct manner without needing to aggregate over observational periods. The REM is therefore especially suited to study the drivers of the development of social interaction over time.

The REM models both *when* a social interaction occurs and *who* will be involved as a function of endogenous and exogenous variables. Temporal dependencies between the events in the relational event history can be introduced into the model by including endogenous variables that refer to characteristics of the past history of events [1]. For example, triadic closure [3] is an endogenous driving mechanism of social interaction in which individuals are more likely to start an interaction with another individual if they have more past interaction partners in common, i.e., "friends of friends become friends" [4]. Exogenous variables refer to any factor outside the history of events that influences social interaction occurrence. For example, homophily, where individuals interact more with others with whom they share one or more individual attributes, such as sex or age, is a well-documented tendency for many social networks [3, 5]. In sum, the REM enables a researcher to study to what extent a combination of endogenous and exogenous variables drive the occurrence, rhythm, and speed of individuals interacting with each other over time.

An important assumption of the basic REM is that the effects on social interaction occurrence are constant over the study period. It is, however, often more plausible that effects change over time. Throughout this paper, we will refer to REM parameters that may change over time as "dynamic," "temporal," or "time-varying." In several articles, the importance of evaluating the assumption of constant effects in REM's has been emphasized (e.g., [4, 6, 7]). First, because the REM assumes that the effects act homogeneously over the course of the observed relational event history, the resulting parameter estimates average away any variation that may be present. Application of a basic REM may thus mask variation of effects over time and inferences drawn on the resulting parameter estimates may be erroneous as a result. Second, time-varying parameters may be intrinsically interesting. Insights in how effects change over time has the potential to progress the understanding of social interaction dynamics.

Studies that relax the assumption that effects are constant over the study period have indeed found evidence for time-varying effects. For example, [6] propose to estimate separate models in which the dependent variable is segregated in discrete time-intervals of interest. In an empirical analysis of the drivers of patient referrals between hospitals, the authors expected daily variations in the effects. Therefore, they estimated seven separate models, one for each day of the week. Results confirmed that drivers of patient referrals between hospitals operated differently for different days of the week.

The approach of [6] is especially suited when time-specific variation of the effects can be expected beforehand based on theory. Unfortunately, since time is only limited accounted for in current social network theories [7], it can be challenging to form theoretically informed hypotheses on time-specific variations in the effects. Moreover, effects in relational event history data may develop irregularly or more smoothly over time. If this is the case, it becomes infeasible to estimate separate models for different time periods. Hence, an approach that does not put any constraints on the development of effects could assist to explore how effects in REMs change over time.

The approach of [8] allows for time-varying regression coefficients in REMs. On synthetic data, the authors illustrated that their model was able to accurately recover both underlying true fixed and time-varying model coefficients. Furthermore, [8] compared the predictive power of a model with time-varying coefficients (the additive Aalen model) and a model with fixed coefficients (the multiplicative Cox model). Results from their prediction experiment on an empirical data set showed that the additive Aalen model significantly outperformed the multiplicative Cox model. These findings further illustrate the importance of testing and accounting for temporal dynamics of the effects in the analysis of empirical relational event history data.

A limitation of the additive Aalen model of [8] for practical use is that the form of the model does not prevent against hazard functions that are estimated to be negative, which are not defined in practice. The moving window approach of [9] for estimating time-varying effects does not have this problem. In this extension of the REM, a moving window slides over the observed relational event history and provides a picture of how the drivers of social interaction develop over time. [9] show in an empirical analysis how the moving window approach can be used to uncover new insights about interaction dynamics. For example, results showed that homophily effects on the probability that employees send each other emails about innovation changed gradually over the course of a year.

The current paper develops an extension of the REM for testing and exploring time-varying effects in relational event history data. First, because of the importance of dynamic network effects, we propose a Bayesian method that tests whether network effects change over time. Such a test is currently missing in the literature. Second, because it is usually not known a priori how well a moving window REM is able to find dynamic network trends, we conduct a simulation study to investigate the accuracy and precision of the methodology. Third, because most theories of social network behavior do not inform researchers on how network effects may vary over time, we propose a data-driven moving window to appropriately balance between accuracy and precision of the moving window REM. Finally, we illustrate the proposed methods in an analysis of the drivers of face-to-face contacts between employees in a workplace [2].

The remainder of this paper is structured as follows. First, we provide a general introduction to the basic REM and describes an extension with time-varying network effects. Subsequently, we introduce a Bayesian approach for testing for dynamic network effects. A simulation study is conducted to evaluate the ability of the test to distinguish between static and dynamic network effects. Next, we introduce the moving window REM, with pre-specified window widths and empirically determined window widths. A simulation study is conducted to study how well the approaches can recover the underlying true time-varying parameters. Subsequently, we describe the methods and results for the illustrative empirical analysis. Finally, we end with a discussion of the research in this article.

## A REM with time-varying network effects

At each observed time point $t$ of the relational event history, the observed sender-receiver pair $(s, r)$ is one out of a set of sender-receiver pairs that can potentially interact. We refer to the set

of sender-receiver pairs $(s, r)$ that can potentially interact at time $t$ as the *risk set*, $\mathcal{R}(t)$. When every actor can be both sender or receiver of the relational events and self-to-self events are excluded, the risk set consists of $N \times (N − 1)$ sender-receiver pairs that can potentially interact, with $N$ referring to the total number of actors in the network.

Each sender-receiver pair $(s, r)$ in the risk set $\mathcal{R}(t)$ occurs with its own rate in the observed event history. We refer to this rate of occurrence at time $t$ for sender-receiver pair $(s, r)$ as the event rate, $\lambda(s, r, t)$. In the REM [1], the event rate is modeled as a log-linear function of endogenous and exogenous variables:

$$\log \lambda(s, r, t) = \sum_{p=1}^{P} \theta_p X_p(s, r, t). \tag{1}$$

Here, $X_p(s, r, t)$ refers to statistic $p = 1, \ldots, P$ for the actor pair $(s, r)$ at time $t$ and $\theta_p$ refers to the model parameter related to statistic $X_p$. The statistics $X_p(s, r, t)$ are numerical representations of the endogenous and exogenous variables in the model.

The event rate determines both the waiting time until the next event and which pair $(s, r)$ is most likely to occur next. Following [1], the waiting time $\Delta t$ until the next event is assumed to follow an exponential distribution:

$$\Delta t \sim \mathrm{Exp}\left( \sum_{(s,r) \in \mathcal{R}(t)} \lambda(s, r, t) \right). \tag{2}$$

The probability to observe the pair $(s, r)$ next at time $t$ follows from the categorical distribution:

$$P((s, r)|t) = \frac{\lambda(s, r, t)}{\sum_{(s,r) \in \mathcal{R}(t)} \lambda(s, r, t)}. \tag{3}$$

Throughout this paper, we consider a REM with dynamic network effects where the rate parameter is defined by:

$$
\begin{aligned}
\log \lambda(s, r, t) = \quad & \theta_{\mathrm{baseline}}(t) X_{\mathrm{baseline}}(s, r, t) \;+ \\
& \theta_{\mathrm{Z.of.sender}}(t) X_{\mathrm{Z.of.sender}}(s, r, t) \;+ \\
& \theta_{\mathrm{difference.in.Z}}(t) X_{\mathrm{difference.in.Z}}(s, r, t) \;+ \\
& \theta_{\mathrm{activity}}(t) X_{\mathrm{activity}}(s, r, t) \;+ \\
& \theta_{\mathrm{inertia}}(t) X_{\mathrm{inertia}}(s, r, t) \;+ \\
& \theta_{\mathrm{transitivity}}(t) X_{\mathrm{transitivity}}(s, r, t)
\end{aligned}
\tag{4}
$$

Here, $X_{\mathrm{baseline}}(s, r, t) = 1$ (i.e., an intercept), $X_{\mathrm{Z.of.sender}}(s, r, t)$ is equal to the value of exogenous variable Z of sender $s$, $Z \sim \mathcal{N}(0, 1)$, $X_{\mathrm{difference.in.Z}}(s, r, t)$ is equal to the absolute difference between the value of Z of sender $s$ and receiver $r$, $X_{\mathrm{activity}}(s, r, t)$ is equal to the standardized outdegree of sender $s$ at time $t$, $X_{\mathrm{inertia}}(s, r, t)$ is equal to the standardized number of past events sent by sender $s$ to receiver $r$ at time $t$, and $X_{\mathrm{transitivity}}(s, r, t)$ is equal to the standardized number of past outgoing two-paths between sender $s$ and receiver $r$ at time $t$. The corresponding model parameters are referred to by $\theta$ and may vary over time.

We define the following four scenarios with time-varying parameters:

**Table 2. Information on the model parameters in the four scenarios for time-varying effects.**

| Effect | Constant | Cyclic | | Gradual | | Mixed |
|---|---|---|---|---|---|---|
| | $\theta$ | *a* | *b* | *c* | *d* | Change |
| Baseline | -8.00 | 0.50 | -8.00 | 1.00 | -8.50 | Cyclic |
| Z of sender | 0.20 | 0.10 | 0.20 | 0.20 | 0.10 | Constant |
| Difference in Z | -0.20 | 0.10 | -0.20 | 0.20 | -0.30 | Constant |
| Activity of sender | 0.10 | 0.05 | 0.10 | 0.10 | 0.05 | Cyclic |
| Inertia | 0.10 | 0.05 | 0.10 | 0.10 | 0.05 | Gradual |
| Transitivity | 0.20 | 0.10 | 0.20 | 0.20 | 0.10 | Gradual |

1. *Constant effects*; in this scenario we assume that the effects of the statistics on the relational event history are constant over time (i.e., they do not change). The corresponding model parameters $\theta$ can be found in Table 2 and are visualized in Fig 1.

2. *Cyclic change*; in this scenario we assume cyclic changes in the effects of the predictors on the relational event history over time. Here, we focus on the cases in which cyclic patterns of change in the data are not necessarily expected beforehand, but are to be detected from the data. Alternatively, when time-specific variation (e.g., weekdays versus weekend days or daytime versus nighttime) can be expected to induce cyclic patterns in the effects of interest, these can be studied with the approach of [6]. To let the model parameters change cyclically over time, we use the following sine functions:

$$a\sin\left(\frac{2\pi}{10000}t\right) - b. \tag{5}$$

The values for *a* and *b* per predictor can be found in Table 2, the resulting model parameters $\theta$ are visualized in Fig 1.

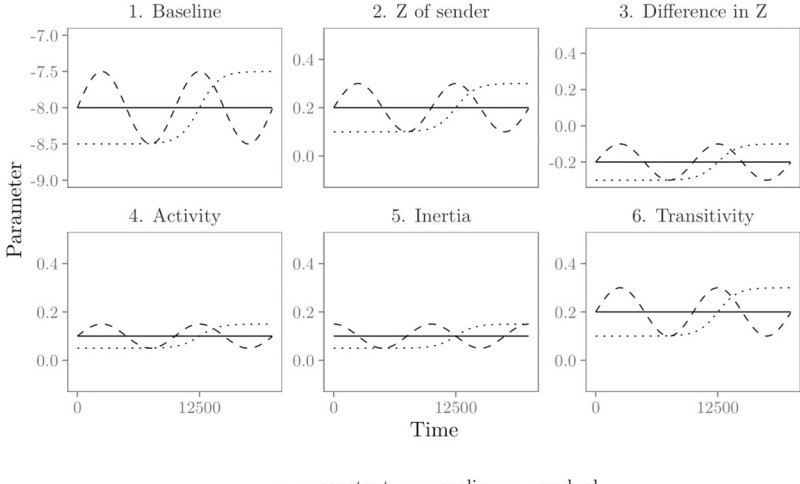

**Fig 1. Parameters in the four scenarios for time-varying effects.** Solid lines show the parameters for the constant effects in the 'constant' and 'mixed' effects scenarios, dashed lines show the parameters for the cyclically changing effects in the 'cyclic' and 'mixed' effects scenarios, and dotted lines show the parameters for the gradually changing effects in the 'gradual' and 'mixed' effects scenarios.

3. *Gradual change*: in this scenario we assume that the effects of the predictors on the relational event history change gradually over time until they stabilize at a "new normal." To let the model parameters change gradually over time we use the following logistic function:

$$\frac{c}{1 + \exp\left[-0.001(t - 12500)\right]} + d \tag{6}$$

The values for $c$ and $d$ per predictor can be found in Table 2, the resulting model parameters $\boldsymbol{\theta}$ are visualized in Fig 1.

4. *Mixed change*: in this scenario we assume that some effects of the predictors on the relational event history stay constant over time, others change cyclically and the remaining effects change gradually. Table 2 shows per predictor the type of change for this scenario in which how the effects change over time is mixed.

These four scenarios for time-varying parameters were chosen to include a baseline scenario with no changes in the effects (constant effects), two scenarios in which effects change over time in a realistic way that may be encountered in empirically collected relational event history data (cyclic and gradual effects) and a scenario in which not every effect changes over time in the same manner (mixed effects). We assume that these four scenarios provide a thorough evaluation of the ability of the methods to capture a diversity of ways in which effects in a REM can vary over time.

For each of these four scenarios, we generate 200 relational event histories with $M = 10000$ events for a network with $N = 20$ actors. Sampling of the events starts at $t = 0$ and continues until 10000 events are reached. At a given time $t$, we sample the waiting time $\Delta t$ until the next event from Eq (2) and the next observed dyad $(s, r)$ from Eq (3). The script files to generate the data and reproduce the analyses performed in this article can be found at https://github.com/mlmeijerink/REHdynamics.

## Testing for time-varying network effects

The first step in an empirical analysis of temporal network data is to test whether it is likely that the effects that drive the interaction between the actors can be assumed constant over the observation period. For this purpose, we formulate two competing hypotheses:

$$H_{\text{static}} : \text{network effects are static} \tag{7}$$

versus

$$H_{\text{dynamic}} : \text{network effects are dynamic.} \tag{8}$$

To evaluate the support in the data for these two competing hypotheses, we divide the observed relational event history into $K$ sub-sequences that are evenly spaced in time, see Fig 2. For example, let $\tau$ define the time of the end of the observation period. Than, for $K = 2$, we obtain two sequences, one with the events observed in the time interval $\left[0, \frac{\tau}{2}\right]$, and one with the events observed in the time interval $\left(\frac{\tau}{2}, \tau\right]$, see Fig 2, upper panel. Subsequently, for each sub-sequence $k = 1, \ldots, K$, the vector with model parameters $\boldsymbol{\theta}_k$ is estimated. The hypotheses in Eqs (7) and (8) can now be re-written as

$$H_{\text{static}} : \boldsymbol{\theta}_1 = \cdots = \boldsymbol{\theta}_k = \cdots = \boldsymbol{\theta}_K \tag{9}$$

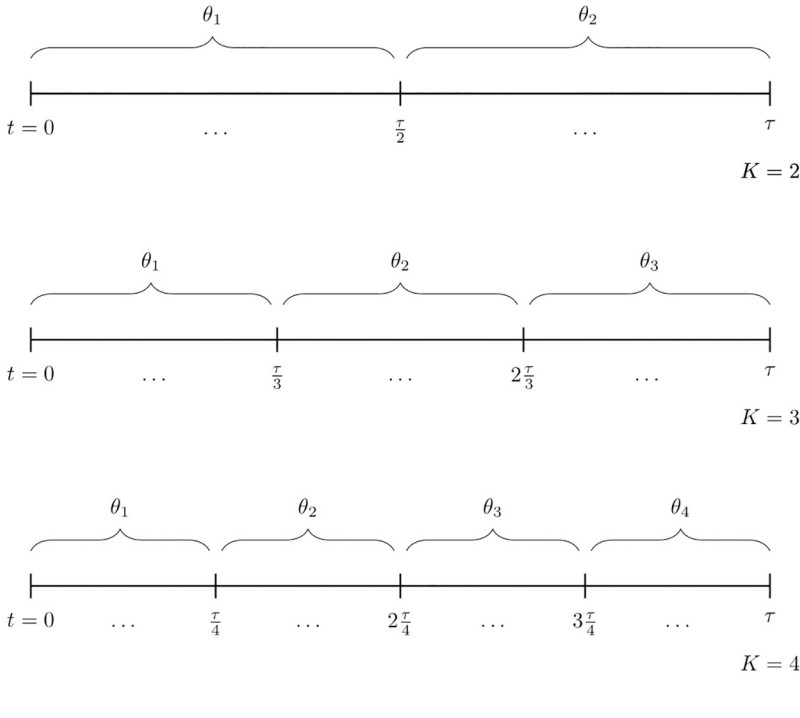

**Fig 2. Illustration of the procedure behind the Bayesian test for time-varying network effects in relational event history data.** The observed relational event history is divided into $K$ sub-sequences that are evenly spaced in time. For each sub-sequence $k = 1, \ldots, K$ the vector of model parameters $\theta_k$ is estimated. The statistical evidence in the data for $H_{\text{static}}$: $\theta_1 = \cdots = \theta_K$ versus $H_{\text{dynamic}}$: not $H_{\text{static}}$ is evaluated by means of the BF.

versus

$$H_{\text{dynamic}} : \text{not } H_{\text{static}}. \tag{10}$$

We propose to compute Bayes factor [10, 11] for the evaluation of the two competing hypotheses in Eqs (9) and (10). In contrast to null hypothesis significance testing, the objective of hypothesis evaluation using Bayes factor is not to arrive at a dichotomous decision on whether a hypothesis is rejected or not, but to determine the probability of the data under one hypothesis versus another hypothesis [12]. For example, a Bayes factor of 10 for the comparison of $H_{\text{static}}$ against $H_{\text{dynamic}}$ indicates that there is 10 times more statistical evidence in the data for the hypothesis that effects are static compared to the hypothesis that effects are dynamic.

Due to prior sensitivity of Bayes factor, we propose to use the multiple population adjusted approximate fractional Bayes factor (from now on abbreviated to BF), which can be computed in an automatic fashion without having to formulate any substantive prior beliefs [13–16]. The BF uses a fraction $b_k$ of the information in the likelihood for each sub-sequence $k$ to construct an implicit default prior [14, 16]. We follow the recommendation in [14], and choose

$$b_k = \frac{1}{K} \times J^* \times \frac{1}{N_k}, \tag{11}$$

where $J^*$ refers to the number of independent constraints in $H_{static}$ (i.e., $J^* = K - 1$), and $N_k$ denotes the number of events in sub-sequence $k$. From [14], it follows that the relative support in the data for $H_{static}$ and $H_{dynamic}$ can be quantified using:

$$\text{BF} = \frac{\int_{\theta \in H_{static}} \mathcal{N}(\boldsymbol{\theta}|\hat{\boldsymbol{\theta}}, \hat{\boldsymbol{\Sigma}}_{\boldsymbol{\theta}})d\boldsymbol{\theta}}{\int_{\theta \in H_{static}} \mathcal{N}(\boldsymbol{\theta}|\boldsymbol{\theta}_B, \hat{\boldsymbol{\Sigma}}_{\boldsymbol{\theta}}^b)d\boldsymbol{\theta}}, \tag{12}$$

that is the ratio of the fit and the complexity of $H_{static}$ relative to $H_{dynamic}$. Here, $\boldsymbol{\theta} = [\boldsymbol{\theta}_1, \ldots, \boldsymbol{\theta}_k, \ldots, \boldsymbol{\theta}_K]$, $\hat{\boldsymbol{\theta}}$ denotes the maximum likelihood estimate of $\boldsymbol{\theta}$, $\hat{\boldsymbol{\Sigma}}_{\boldsymbol{\theta}}$ denotes the corresponding co-variance matrix, $\boldsymbol{\theta}_B$ is the adjusted mean of the prior distribution, here, 0, i.e., a value of $\boldsymbol{\theta}$ on the boundary of all hypotheses under investigation [15], and $\hat{\boldsymbol{\Sigma}}_{\boldsymbol{\theta}}^b$ denotes the covariance matrix of the prior distribution of $\boldsymbol{\theta}$, which is based on a fraction $b$ of the information in the data, where $b = [b_1, \ldots, b_k, \ldots, b_K]$. Maximum likelihood estimates $\hat{\boldsymbol{\theta}}_k$, and, for each sub-sequence $k$, corresponding covariance matrix $\hat{\boldsymbol{\Sigma}}_{\boldsymbol{\theta}_k}$, are easily obtained from R software packages tailored for relational event modeling, and $\hat{\boldsymbol{\Sigma}}_{\boldsymbol{\theta}_k}^{b_k}$ is computed as $\frac{\hat{\boldsymbol{\Sigma}}_{\boldsymbol{\theta}_k}}{b_k}$. The BF factor shows consistent behavior, which implies that the evidence for the true hypothesis will increase to infinity as the sample size grows [11, 14].

We recommend to compute the BF for increasing K from 2 to 10 (or more in the case of unclear results). By considering multiple values for K, we get insights into the time scale of the dynamic behavior of the network effects. Further, note that as $K$ increases, the number of parameters under $H_{dynamic}$ increases and thus the evidence for $H_{static}$ will increase if the data suggest that the static REM fits bests.

In order to evaluate the ability of the proposed Bayesian test to prefer the true model ($H_{static}$ or $H_{dynamic}$), we conduct a simulation study. We compute the BF factor for the evaluation of $H_{static}$ versus $H_{dynamic}$ with $K = 2, \ldots 10$ for the 200 generated relational event histories in the four time-varying effects scenarios. Results in Fig 3 show the mean log BF and its 95%

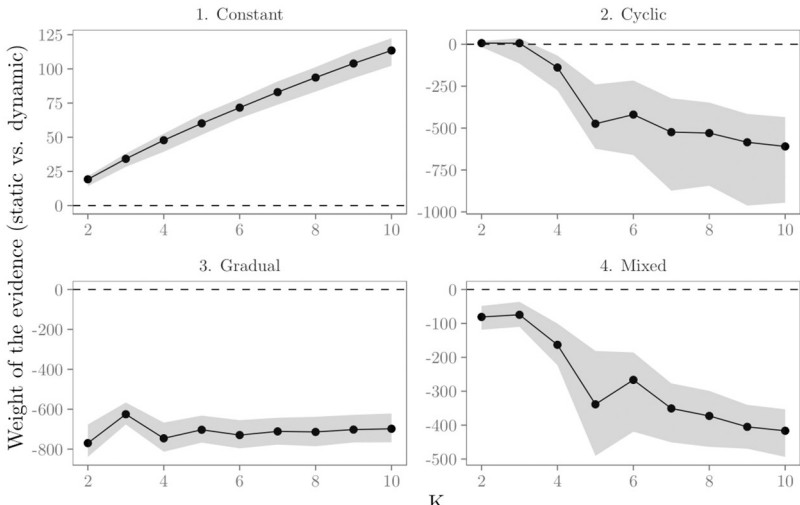

**Fig 3. Results for the numerical evaluation of the Bayesian test for time-varying parameters.** The y-axis shows the size of the log BF, i.e., the weight of the evidence for the static hypothesis versus the dynamic hypothesis. Black dots and solid lines show per effect scenario the mean log BF with increasing K. The gray area shows the 95% sampling distribution of the log BF.

sampling distribution across the increasing values of $K$. If the log BF is larger than zero, this implies most evidence in the data for $H_{static}$. If the log BF is smaller than zero, most evidence in the data is in favor of $H_{dynamic}$.

The upper left panel in Fig 3 shows that in the 'constant' effects scenario, the evidence is on average largest for the static hypothesis. In fact, the log BF is in favor of the true model ($H_{static}$) for all $K = 2, \ldots, 10$ in all 200 generated data sets, i.e., in all data sets the results of the test indicate that the relational event history can be best analyzed with a static REM. These results are what is wanted because the effects in the relational event histories don't change over time in this scenario. Furthermore, the results show that the evidence increases with increasing $K$. This shows that the BF functions as an Occam's razor that penalizes larger models. Results in the other panels of Fig 3 are indicative of time-varying parameters: for almost all $K$ (except $K = 2$ and $K = 3$ in the "cyclic" scenario) the evidence in the data is largest for $H_{dynamic}$. These results are what one wants, because these three scenarios include time-varying parameters. Hence, based on these results, the proposed Bayesian test for time-varying parameters seems to be able to accurately distinguish between data with static and dynamic effects and to provide guidance as to whether the relational event history can best be analyzed with a static REM or whether a more dynamic approach is required.

## Exploring time-varying network effects

### Moving window REM

A few studies have explored time-varying network effects in relational event history data by fitting the model on different observational periods [6, 9, 17]. Here, we propose to use a moving window REM for exploring time-varying network effects. Algorithm 1 describes the steps in fitting a moving window REM. In summary, a REM is fitted on the sub-sequence of events that fall within a pre-specified time-interval or 'window.' By sliding this window over the relational event sequence and estimating the REM for each corresponding sub-sequence of events, a view of the trend in the parameter estimates over time is obtained.

**Algorithm 1**: Moving window REM

```
input: A relational event sequence with M events between t = 0 and t =
       τ.
1 Set the window width ℓ;
2 Set the proportion of overlap between subsequent windows π;
3 Start at the first window w = 1;
4 while (ℓ + (w − 1)πℓ) ≤ τ do
5    Select the set of events observed in the time-interval between [(w
     − 1)πℓ, ℓ + (w − 1)πℓ];
6    Compute the statistics X(s, r, t) for the selected events in the
     window w;
7    Estimate the vector of model parameters θw for the selected events
     in the window w;
8    Continue at the next window w = w + 1;
output: Vector of estimated model parameters θw for each window w.
```

As Algorithm 1 states, the researcher has to define the window width and the proportion of overlap between subsequent windows. A higher number of events that overlap between subsequent windows results in greater smoothness of the results. Numerous factors play a role in determining the window width, including the following:

1. Social theory or field knowledge. In certain situations, social theory or field knowledge can suggest how fast social interaction behavior changes over time. In these situations, the window width should correspond to the expected rate of change of the effects, i.e., a narrow

(wide) window should be used when theory dictates that interaction behavior is highly (hardly) dynamic.

2. Research question. A window width should be chosen corresponding to the research question at hand, e.g., are researchers interested in daily, monthly, or annual dynamics?

3. Resolution of the data. In certain situations, the possible window widths may be limited by the resolution of the data, e.g., whether the time of the events is available in seconds, hours, or days.

4. Precision/accuracy trade-off. A window should be wide enough so that enough events fall within each window to estimate effects with sufficient precision. At the same time, a window that is too wide may average out small or moderate changes in the effects, resulting in loss of accuracy of the estimates. Unfortunately, studies into the power, accuracy and precision of REMs are currently limited. The results of one study suggest 100 events per actor to achieve good power [18].

## A data-based method for balancing precision and accuracy in moving window REMs

One challenge of the moving window approach is to determine the window width that can best capture how the effects on social interaction develop over time. The moving window REM uses a fixed window width and slides that window across the entire event sequence. However, in certain situations the time-varying parameters may change quite fast and quite a lot in some parts of the observation period and a lot less in in other parts (e.g., see the "gradual" time-varying effects scenario). In these situations, an optimal precision/accuracy trade-off can only be achieved by allowing the window widths to themselves vary over time. Unfortunately, considering that most research on relational event histories is still fairly exploratory, there is little theory yet to guide us how to set the window width in which part of the observation period. For this reason, we propose a method to empirically determine the window width based on the observed event history, where a narrow (wide) window is used during phases when the data show important (hardly any) changes in social interaction behavior, balancing precision and accuracy of the parameter estimates.

The steps in the procedure for the data-driven moving window REM are described in Algorithm 2. We make use of the BF to determine the window width around a given time point. Due to the Occam's razor, the BF is very suitable to optimize the window width around a given time point by balancing between precision and accuracy. More events will be preferred when possible, and fewer events when necessary. First, at a given time point $t$, a small window width is proposed. We evaluate if the effects around $t$ change during the proposed window width by computing the BF for the evaluation of $H_{\text{static}}$ (Eq 9) versus $H_{\text{dynamic}}$ (Eq 10) with $K = 3$ for the events in this window. If the log BF is larger than zero, there is more evidence in the data for the static hypothesis, i.e., the BF indicates that the effects do not change during the proposed window around $t$. Subsequently, we repeatedly increase the window width, i.e., repeatedly select more events to estimate the effects around $t$. For each increased window width, we evaluate if the effects change during the window around $t$ by computing the BF with $K = 3$. As long as the log BF is larger than zero, we can conclude that the effects during the window do not change. In the algorithm, we implement a stopping rule to increase its computational efficiency. That is, we stop increasing the window width around $t$ when the log BF is smaller than $\log \frac{1}{10}$, i.e., there is ten times more evidence in the data in favor of the dynamic hypothesis. When this happens, we set the window width around $t$ equal to the window width for which

BF was maximum, i.e., there was most evidence in the data for static effects. This allows us to estimate the vector of model parameters at $t$ with more events when possible (when effects do not change) and fewer events when necessary (when effects change), hence with maximum precision and accuracy. The algorithm for the data-driven moving window REM only requires a *minimum* window width to be set.

**Algorithm 2**: Data-driven moving window REM

**input**: A relational event sequence with $M$ events between $t = 0$ and $t = \tau$.

1 Set the minimum window width $\ell_{\min}$;
2 Define the set of time points $T$ around which an optimal window width will be determined as follows: $T = \{\frac{1}{2}\ell_{\min},\ \frac{1}{2}\ell_{\min} + \frac{1}{3}\ell_{\min},\ \frac{1}{2}\ell_{\min} + \frac{2}{3}\ell_{\min},\ \ldots\}$;
3 **for** $t \in T$ **do**
4 Set the window width around $t$ equal to $\ell = \ell_{\min}$ around $t$;
5 Select the set of events observed in the time-interval between $[t - \ell,\ t + \ell]$;
6 Compute the statistics $X(s,\ r,\ t)$ for the selected events in the window around $t$;
7 Compute the BF with $K = 3$ for the selected events in the window around $t$;
8 **if** $\log BF \geq \log \frac{1}{10}$ **then**
9 Increase $\ell = \ell + \frac{2}{3}\ell_{\min}$;
10 Start again at line 5;
11 **else**
12 Set the window width around $t$ equal to the $\ell$ for which BF was maximum;
13 Estimate the vector of model parameters $\boldsymbol{\theta}_t$ with the events in the window;

**output**: Vector of estimated model parameters $\boldsymbol{\theta}_t$ for each time point $t \in T$.

## Numerical evaluation

We conduct a simulation study to assess the accuracy and precision of the moving window REM with fixed (Algorithm 1) and data-driven (Algorithm 2) window widths. First, we fit a "static" REM to the 200 generated relational event histories in the four time-varying effect scenarios. Second, we fit a moving window REM with fixed window widths. To study the accuracy and precision across window widths, we apply three different window widths (1000$t$/ 'small', 2000$t$/'medium', and 4000$t$/'large'). We slide the windows such that they have a two-thirds overlap with the previous window. Finally, we fit a data-driven moving REM. The minimum width is set equal to 1000$t$. Statistics are computed with the R package REMSTATS, estimation of the model parameters is done with the R package REMSTIMATE. Both these R package are available for download at https://github.com/TilburgNetworkGroup.

Figs 4–6 show the results of the numerical evaluation for the 'transitivity' effect. Results for the other effects show similar patterns, see S1 File. Furthermore, Fig 7 shows the average data-based window width per time point as determined by the data-driven moving window REM.

Fig 4 shows the average estimated model parameter over time. First, results in the top row of Fig 4 show that the "static REM" averages out any time-variation that is present. Furthermore, results in Fig 4 show that the moving window REM, both with fixed and data-driven window widths, is able to provide an informative view of the underlying trend in parameters over time. The accuracy and precision of this view, however, depend on the window width and the extent and kind of time-variation of the parameters, as shown in more detail in Figs 5 and 6.

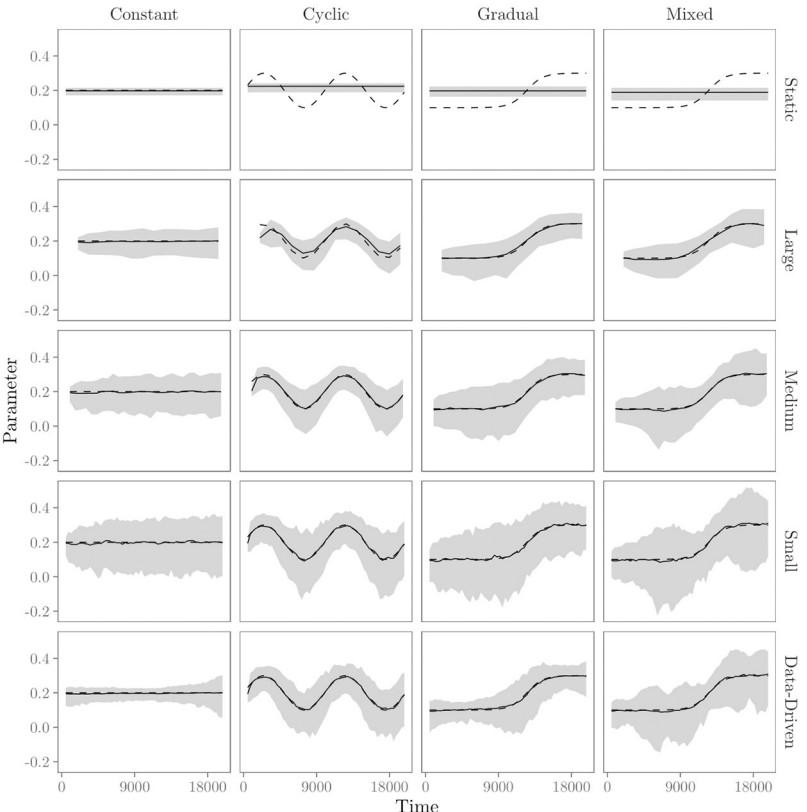

**Fig 4. Results from the evaluation of the (moving window) REM for the 'transitivity' effect.** Rows show results for estimation of the 'transitivity' effect with the 'static' REM, large (4000*t*), medium (2000*t*), small (1000*t*), and data-based window widths, respectively. Columns show results for estimation of the 'transitivity' effect in the four time-varying effects scenarios. Solid lines represent the mean estimated parameters over 200 datasets over time. The gray area represents the range with 95% of the estimates for the 200 datasets. Dashed lines represent the parameters used for data generation.

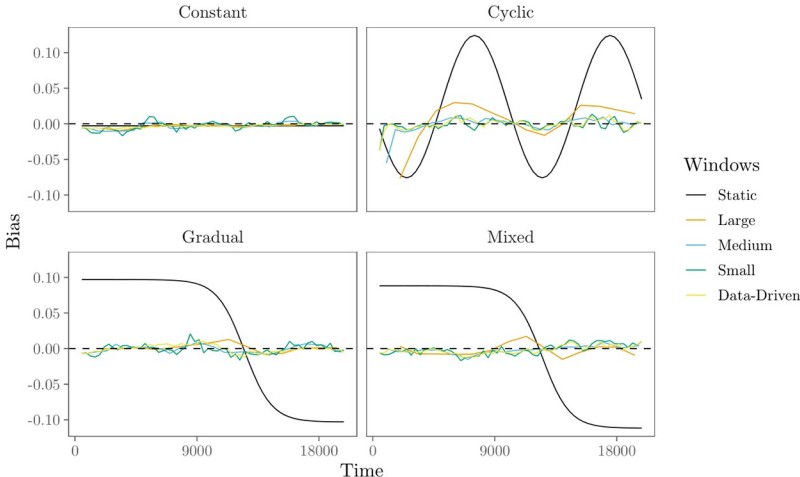

**Fig 5. Bias for the 'transitivity' effect in the evaluation of the moving window REM.** Panels refer to the four time-varying effect scenarios. Solid lines represent the bias of the parameter estimates over time, with colors representing estimation with large (4000*t*), medium (2000*t*), small (1000*t*) and data-based window widths.

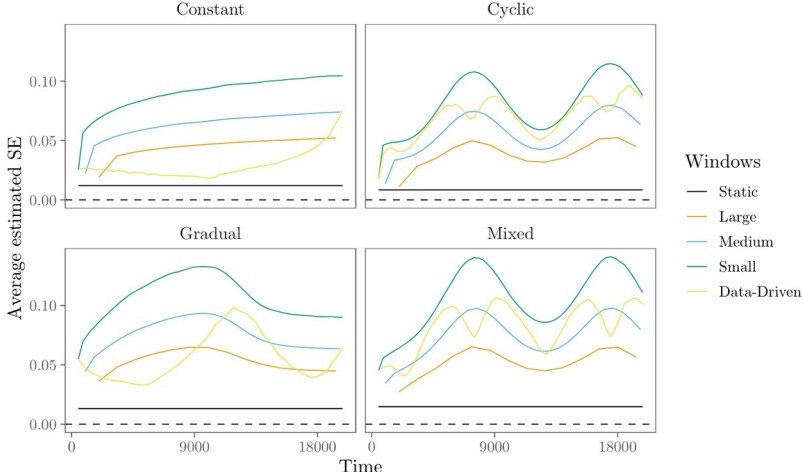

**Fig 6. Average estimated standard error (SE) for the 'transitivity' effect in the evaluation of the moving window REM.** Panels refer to the four time-varying effect scenarios. Solid lines represent the average estimated SE of the parameter estimates over time, with colors representing estimation with large (4000*t*), medium (2000*t*), small (1000*t*) and data-based window widths.

Fig 5 provides some insights in the accuracy of the moving window REM. We use the bias of the estimated parameters as a measure of accuracy. The bias quantifies how well the true underlying parameter $\theta$ is quantified by the estimator on average. For time *t*, it is calculated as

$$\text{bias}(t) = \frac{1}{n_{\text{sim}}} \sum_{i=1}^{n_{\text{sim}}} \hat{\theta}_i(t) - \theta(t), \tag{13}$$

where $n_{\text{sim}}$ denotes the number of simulated datasets—here 200, $\hat{\theta}_i(t)$ denotes the estimated parameter in dataset *i* at time *t* and $\theta(t)$ denotes the true parameter at time *t*. We highlight three interesting results that follow from Fig 5. First, the results for the data generated in the

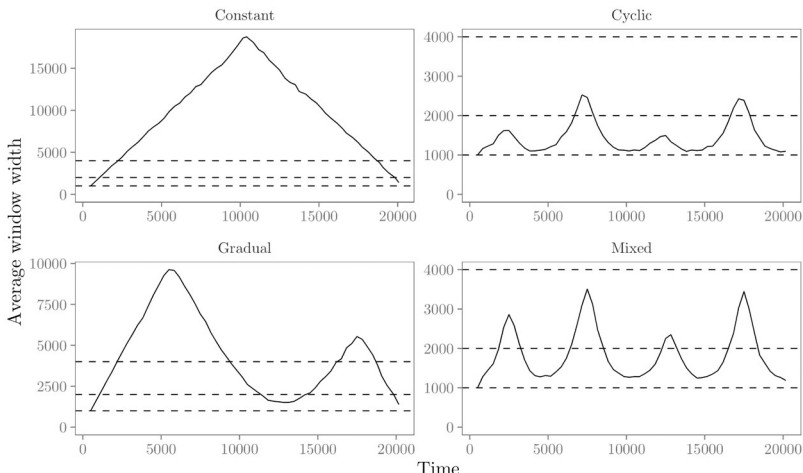

**Fig 7. Average data-driven window width found in the data-driven moving window REM for the four time-varying effect scenarios.** Horizontal dashed lines refer to the different fixed window widths evaluated for the moving window REM, i.e., large (4000*t*), medium (2000*t*), small (1000*t*).

'constant' scenario (upper left panel) indicate that the bias of the estimates in the moving window REM, both with fixed and data-driven window widths, is generally very low, -the moving window REM is able to estimate effects with good accuracy. Second, results indicate that the bias of the parameter estimates can become quite large if effects do change and the window widths are too large to capture that change. For example, the upper right panel of Fig 5 shows that the largest window widths are clearly too large to accurately estimate the highly dynamic transitivity effect in the data generated in the 'cyclic' scenario. Third, since smaller window widths signify greater model flexibility, it follows that bias is lower for smaller window widths. For all three time-varying effects scenarios, the bias is estimated to be reasonable small for medium, small and data-driven window widths. In sum, when effects are quite stable, bias is low for all window widths. When effects are highly dynamic, however, smaller window widths clearly outperform larger window widths by having lower bias. The algorithm for data-driven window widths enables us to find out when effects are highly dynamic, and thus window widths should be small.

Fig 6 provides insight into the precision of the moving window REM. We use the average estimated standard error (SE) of the parameters as a measure of precision. Following [19], it is calculated as

$$\text{average estimated } \text{SE}(t) = \sqrt{\frac{1}{n_{\text{sim}}} \sum_{i=1}^{n_{\text{sim}}} \widehat{\text{Var}}(\hat{\theta}_i(t))}.$$ (14)

As expected, Fig 6 shows that the average estimated SE is larger for smaller windows in all four scenarios. While smaller windows may provide higher accuracy of the estimated parameters when effects are highly dynamic, this comes at the costs of lower precision since fewer events are contained inside each window. The upper left panel of Fig 6 shows that in the 'constant' scenario, the average estimated SE increases over time, even though the effects do not change. This is most likely due to the increased variability in the statistics over time, as the network that is observed grows. A similar pattern is also observed in the other time-varying effect scenarios. The upper right panel of Fig 6 shows the average estimated SE in the 'cyclic' scenario. As can be seen, the trend in size of the average estimated SE mirrors the trend of the parameters over time. This is mainly due to the fact that, when the baseline and other effects in the model are smaller, fewer events are generated/observed, leading to an increase in the estimated SE. A similar pattern is also observed in the 'gradual' and 'mixed' scenarios. Finally, from Fig 5 we could conclude that when effects were highly dynamic the bias was reasonably low for medium, small and data-driven window widths. The advantage of the data-driven window widths was that they enable us to find out when effects are highly dynamics and small(er) window widths are therefore required to estimate effects with enough accuracy. From Fig 6, we see another advantage from the data-driven window widths: they allow us to find out when effects are stable enough to increase the window widths to estimate effects with greater precision (smaller SE). This becomes especially apparent in the "gradual" change scenario, depicted in the lower left panel of Fig 6. Here, the SE is considerably lower for the data-driven window widths compared to the small and medium fixed window widths for the first half and towards the end of the study period, i.e., when effects are more stable.

In sum, results from the numerical evaluation show that the moving window REM is able to provide a clear view of the trend in parameters over time. However, the window width influences the accuracy and precision of this view, depending on how much the parameters vary over time. Results further show that we can use the proposed algorithm for data-driven window widths to find out when effects are highly dynamic and we should decrease the window

widths to estimate effects with greater accuracy, and when effects are stable enough to increase the window widths to estimate effects with greater precision.

## Application: Time-varying network effects in workplace contacts

With the methods in place, we now perform an illustrative empirical analysis to demonstrate 1) the use of the Bayesian test for time-varying network effects and 2) the moving window REM with fixed and data-based window widths for exploring time-varying network effects. In particular, we focus on how past interaction behaviors (i.e., endogenous mechanisms) affect future contacts between the employees and how these effects change over time. The script files to reproduce the analyses can be found at https://github.com/mlmeijerink/REHdynamics.

### Data

The data set contains the face-to-face contacts between 232 employees of an organization in France, measured during a 2 week time period in 2015 [2]. These face-to-face contacts were measured with close-range proximity sensors. Following [2], a contact between two employees is defined as "a set of successive time-windows of 20 seconds during which the individuals are detected in contact, while they are not in the preceding nor in the next 20 second time window." We formally represent a relational event between two employees as the triplet $(s, r, t)$, where $s$ and $r$ refer to the employees who are in contact and $t$ refers to the start time of the face-to-face contact in seconds since onset of observation. The events do not distinguish between a sending and receiving individual, i.e., the relational events are undirected. The first ten events in the sequence are shown in Table 1. In total, 33751 relational events are observed over the course of the study period. The top panel in Fig 8 shows the distribution of events over time. The number of events per day ranged from 1778 to 5905 with a mean of 3375 (SD = 1166). In the analysis, idle periods, such as non-working hours and weekends, were discarded from the data. Between events that have a same timestamp, a time difference is induced such that these events are evenly spaced in time between the current time unit and the next time unit. The risk set consists of every undirected employee pair that can potentially interact, i.e., $\frac{232 \times 231}{2} = 26796$ pairs. Across the entire relational event history, the number of events per employee ranged from 0 to 1147 with a mean of 291 (SD = 206). The majority of actors (217, 94%) were involved in at least one event during the study period. The number of events per employee pair ranged from 0 to 506 with a mean of 1 (SD = 8) and there were 4274 employee

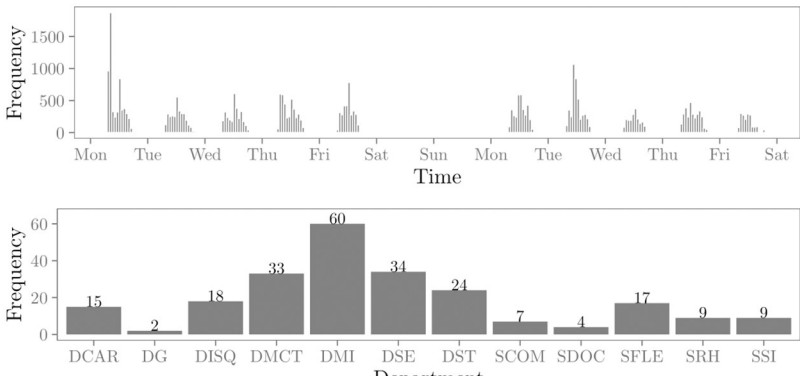

**Fig 8. Descriptive information of the empirical data.** The top panel shows the frequency distribution of the events in the relational event sequence over time. The bottom panel shows the frequency distribution of the employees over the departments.

pairs (16% of the risk set) with at least one event during the study period. Besides the face-to-face contacts, information on the departments in which the employees work is available. The bottom panel in Fig 8 shows the frequency distribution of study participants over the departments.

## Model specification

The following describes the statistics that are used to model the rate (see Eq 1) at which the employee pairs start face-to-face interactions.

**Intercept.** Firstly, an intercept is included in the model to capture the baseline tendency for interaction in the employee network. The statistic $X_{\mathrm{intercept}}(s, r, t)$ is equal to 1 for all $(s, r, t)$. The corresponding effect $\beta_{\mathrm{intercept}}$ refers to the log-inverse of the average number of events per time unit (here, seconds) for an employee pair that scores zero on all other statistics in the model.

**Same department.** Previous research has shown that the number of contacts between employees is strongly influenced by the departmental structure of the organization [20]. Since employees who work in the same department are likely to have a greater opportunity to interact, the statistic $X_{\mathrm{same.department}}(s, r, t)$ is included in the model to capture whether the employees of the pair $(s, r)$ work in the same department (1 = yes, 0 = no). A positive effect $\beta_{\mathrm{same.department}}$ implies that employees who work in the same department start future interactions with a higher event rate with each other compared to employees who work in different departments.

**Recency.** A previous REM analysis of email communication between employees of an organization showed that individuals were more likely to send an email if the last email sent was more recent [9]. Here, we are interested in the question whether such recency effects transfer to face-to-face interactions. Moreover, previous research into the validity of using sensor-based measures of face-to-face interactions has shown that merging interactions that occurred close to reach other in time improved the accuracy [21]. It is possible that individuals physically move away from each other during a face-to-face interaction in such a way that a longer interaction is recorded as two or multiple shorter interactions by the sensors. Hence, the speed of interaction is possibly confounded by this specific source of measurement error. By including a recency effect, we can control for this in the estimated sizes of the other effects in the model. Since we have undirected events, we include an effect to control for the effect of how recently the employee pair $(s, r)$ interacted last. Let $\tau(s, r)$ refer to the time of the most recent event between the employee pair $(s, r)$, then

$$X_{\mathrm{recency}}(s, r, t) = \frac{1}{(t - \tau(s, r)) + 1}.$$

A positive effect $\beta_{\mathrm{recency}}$ implies that employee pairs who interacted more recently tend to engage in future interactions at a higher rate than employee pairs whose last interaction was less recent.

**Inertia.** Inertia refers to the tendency of individuals to repeat past interactions, or the tendency of "past contact to become future contacts" [4]. [4] suggest that, following general theories of social networks, inertia is an important predictor of social interaction occurrence. Previous research on communication between employees has repeatedly found inertia to positively predict the event rate [18, 22–24]. Hence, we may expect that inertia plays an important role in our data as well. We are especially interested in how the effect of inertia develops over time in the employee network. The statistic $X_{\mathrm{inertia}}(s, r, t)$ is based on a count of past $(s, r)$ events before time $t$. Let $m = 1, \ldots M$ refer to the $m$th event in the relational event history and

let $(s_m, r_m)$ and $t_m$ refer to the employee pair and time of the $m$th event, respectively, then

$$X_{\text{inertia}}(s, r, t) = \sum_{m=1}^{M} I(t_m < t \wedge \{s_m, r_m\} = \{s, r\})).$$

To ensure that the statistic is well-bounded (e.g., see [18, 25]), we standardize the statistic per time point $t$ by subtracting the mean of $X_{\text{inertia}}(t)$ from $X_{\text{inertia}}(s, r, t)$ and subsequently divide by the standard deviation of $X_{\text{inertia}}(t)$. A positive effect $\beta_{\text{inertia}}$ implies that employee pairs who have interacted more with each other in the past tend to engage in future interactions at a higher rate than employee pairs who have interacted less with each other in the past.

**Triadic closure.** Triadic closure refers to the tendency of 'friends of friends to become friends' [4]. [4] suggest that, following general theories of social networks, triadic closure is an important predictor of social interaction occurrence. Previous research on communication between employees has repeatedly found triadic closure to positively predict the interaction rate [18, 22–24]. Hence, we may expect that triadic closure plays an important role in our data set as well. We are especially interested in how the effect of triadic closure develops over time in the employee network. The statistic $X_{\text{triadic.closure}}(s, r, t)$ is based on a count of the past interactions with employees $h$ that employees $s$ and $r$ both interacted with before time $t$: Let $A$ refer to the set of employees in the network, then

$$X_{\text{triadic.closure}}(s, r, t) = \sum_{h \in A} \min \left\{ \sum_{m=1}^{M} I(t_m < t \wedge \{s_m, h_m\} = \{s, h\}), \right.$$

$$\left. \sum_{m=1}^{M} I(t_m < t \wedge \{r_m, h_m\} = \{r, h\}) \right\}.$$

We standardize the triadic closure statistic in the same manner as the inertia statistic. A positive effect $\beta_{\text{triadic.closure}}$ implies that the rate of interaction increases as employees have more common past interaction partners.

## Testing for time-varying network effects

We first test for time-varying network effects. Results in Fig 9 show that the BF indicates more evidence in the data for the hypothesis that the effects change over the course of the relational event sequence rather than remaining constant over time. This holds for every number of sub-

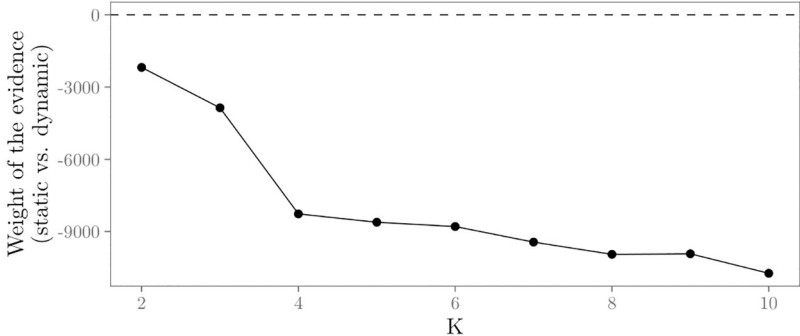

**Fig 9. Results for the tests for time-varying network effects in the empirical data.** Weight of the evidence (log BF) for the static vs. dynamic hypothesis with $K = 2, \ldots, 10$ for the relational event history with face-to-face contacts in the workplace.

sequences $K = 2, \ldots, 10$. Hence, these results indicate the need to a dynamic analysis of the face-to-face interactions between these employees.

## Exploring time-varying network effects

We implement the required dynamic model by means of a moving window REM; this allows us to study the time variation of the effects that influence employee interaction. We apply two different fixed window widths to explore the time-varying network effects: 6 hours and 2 hours. The number of events within the windows range from 375 to 4660 (mean = 1632, SD = 781) and 11 to 2877 (mean = 555, SD = 423), respectively. We also apply the algorithm for data-driven window widths (see Algorithm 2) with a minimum window width of 1 hour.

The results for the two moving window REM's with fixed windows widths and the REM with data-driven window widths are shown in Fig 10. These results show that the largest window (6h) shows the general trend of how the effects develop over time, but it does not pick up many nuances in them. The results suggest that some variations in the effects during the day exist. The smallest window (2h) show these daily variations in more detail, informing us about the magnitude in change of the effects during the day. The results of the data-driven window widths are mostly comparable to the results of the 2h window widths. For most time points, we see a fraction more detail for the data-driven window widths compared to the 2h window widths. For other windows, we see a fraction more precision, i.e., smaller standard errors, for the data-driven window widths compared to the 2h window widths. These results suggest that interaction patterns in the respective workplace are highly dynamic over the course of the study period (changing every $\frac{1}{3} \times 60 = 20$ minutes) and longer periods of stability of the effects do rarely exist.

Overall, results from the analysis with the moving window REM seem to suggest a basic level of importance of the effects throughout the study period. The smaller window width informs us that the general trends over time do not tell us the whole story. All effects show some variation in strength during the working days, through patterns that seem to repeat themselves for most working days. The baseline rate of interaction seems to follow the same pattern every day, with less events during the beginning and the end of the working day and a small drop in the baseline rate around lunch. The results show some evidence that, aside from a baseline importance of the effect of transitivity throughout the study period, its effect on the event rate increases during the day and then resets again at the beginning of a new day. This pattern is especially observed on the first few days. Furthermore, there seems to be a drop in the importance of the effect around noon. Working in the same department has a strong positive effect on the event rate throughout the study period. This effect seems to become an even more important predictor of interaction towards the end of each working day. For recency, it seems that the effect increases somewhat in strength on the third day and then essentially stabilizes. Within these fairly stable period there are several times where recency is higher for awhile. This may be due to a number of reasons, for example because of tasks performed in teams or project work. When these periods of relatively high recency occur, they seem to be concentrated around the end of the working day. The effect of inertia starts strong at the beginning of the study period, but decreases until the fourth day, after which it slowly increases again. This may be due to external influences, for example the end of a large project and the beginning of a new one. Throughout the days, inertia seems to be relatively more important in between the beginning of a working day and noon, and in between noon and the ending of the working day—i.e., the periods during the day when working in the same department was less important. This may point towards employees repeatedly working together on projects, regardless of the department they are in. Overall, it seems to be the case that around noon the

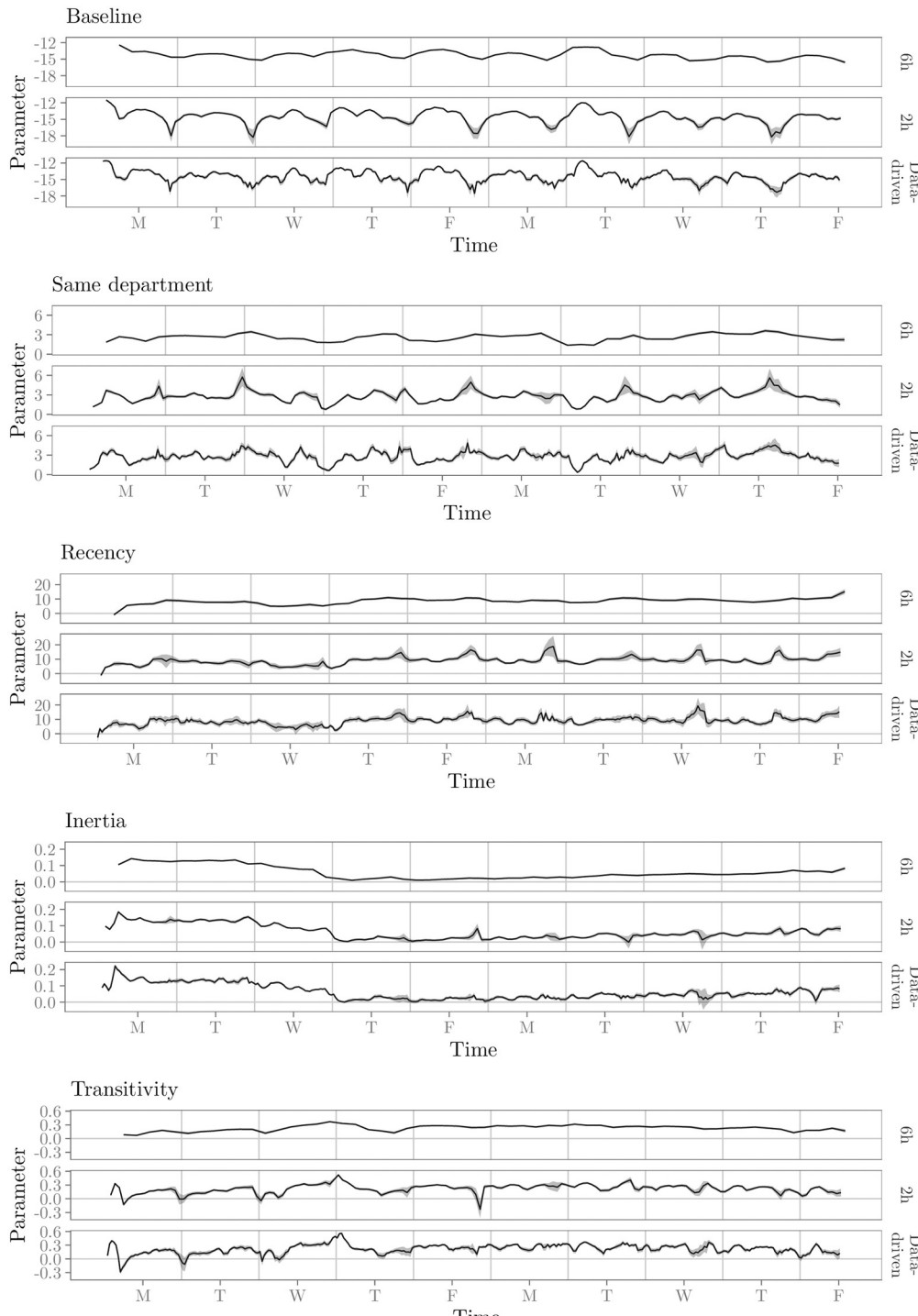

**Fig 10. Results for the moving window REM analysis of the relational event sequence with face-to-face contacts in the workplace, both with fixed and data-driven window widths.**

pattern of work-related interaction may be broken up by the lunch break and, potentially, social interaction with other employees. However, the data set lack this type of contextual information to corroborate this explanation.

## Discussion

The current research proposed three methods to progress the study of temporal effect dynamics in relational event history data. First, we proposed a Bayesian approach to test *if* effects truly change over time. Results showed that the approach is able to provide guidance on whether effects barely change (and a static REM can be applied) or whether effects change considerably (and a dynamic approach is required for the analysis). Second, we argued that the moving window approach enables us to study *how* effects in relational event history data develop over time. Results of a simulation study provided a proof of concept for the moving window approach. The moving window approach was able to recover the time-varying regression coefficients and provide a clear picture of how effects change over time. The accuracy and precision of this picture depend on the window width, with narrower windows resulting in greater accuracy at the cost of lower precision. Finally, since it can be challenging to determine the width of the window for the moving window REM, we propose an algorithm that finds flexible window widths based on the empirical data. The algorithm makes windows wider when effects are stable and narrower when they are dynamic. Results from the simulation study showed that the flexible windows lead to greater *precision* for time intervals in which effects were hardly changing and greater *accuracy* for time intervals in which effects did change greatly over time.

This was also highlighted in the empirical example. Wide fixed windows were able to uncover broad trends, but clearly lacked detail. Making the windows narrower increased this detail, but frequently led to windows that did not have enough observations in them to be sufficiently precise. This is fixed by the data-driven method with flexible window widths, although it will rarely be possible to precisely uncover dynamics in time periods with inherently few events anyway.

The illustrative analysis shows us that the models can indeed retrieve how the dynamics of social interactions change over time. The explanation of what is causing such changes may require additional data. In the empirical example, it appears that there is a lunch effect. To establish this with more certainty, one would like data about what indeed happens in this organization around noon. Do employees lunch together in a cafeteria? If so, it makes sense that interaction may then be driven by other factors than during the regular working hours. Or do the employees lunch with their own team? Or does the organization offer a different activity at noon's? Similarly, we noted that recency is more important during short times (say, periods of 1–2 hours). It would be insightful to know what happened: did colleagues get together to jointly solve work-related problems in those periods? If so, that would (partly) explain the increased recency during those times. There may be other reasons for these effects as well. A dynamic REM approach that allows effects to vary over time is most informative when additional data are available to provide the context within which the interaction take place. Such data may often not be relational event type data and may not always be collected by a researcher who collects data that will be fed into a REM analysis. Our research, however, suggests that such additional data may be highly useful. For employees in an organization, a researcher would like to know at what times formal meetings are scheduled and who is supposed to attend. We would like to know how the work day is organized and which routines are built in (cf., [26]). Moreover, in the current analysis we focus on modeling the predictors of face-to-face interactions between employees in the office space. In many organizations,

employees interact via different modes of communication, e.g., face-to-face interactions, e-mail messages, phone calls, etc. In addition, employees may develop social relationships outside of the office space. All these factors are likely to affect the patterns of observed face-to-face interactions between employees in the office space. In the current analysis, we are restricted due to the limited available information in the data. When such information would be available, however, it is recommended to consider it in the configuration of the predictor variables to obtain a more detailed understanding of the dynamics of employees' face-to-face interactions (e.g., see [27]).

In the last decade, several approaches for modeling relational event data have been introduced (e.g., [1, 8, 22, 28, 29]). Generally, differences between these approaches are small and affect mostly the interpretation of the estimated model parameters [30]. Therefore, while the current research focused on the REM framework [1], we expect that the approach can generalize to other approaches for modeling relational events.

## Conclusion

This paper has provided a tool set for testing for and exploring time-varying network effects in relational event history data. These methods enable researchers to gain insights into how driving mechanisms of social interactions develop over time; when their effects increase, decrease, or remain stable, when effects kick in, how long effects last, et cetera. In the social sciences, there is a dearth of truly time-sensitive theory. As a result, researchers have little guidance in theorizing about *when* events happen, *for how long*, and *what makes events happen* at some points in time but less so at others? We believe that empirical findings by models like the REM can inform the development of time-sensitive theory in the long run. In the current paper, we have suggested a way to make REMs even more informative, by acknowledging that the drivers of social interaction are unlikely to remain constant over time. If we find that organizational routines such as joint lunches break the interaction routines from the first half of the work day, this can inform more nuanced theory building to help us understand how time-specific institutions affect our work interactions. Similarly, it might help us understand how some disruptions (such as routinized lunch time) do not structurally impact employee interaction patterns (employees will likely "continue where they left off" after lunchtime is over), whereas other kinds of disruptions (e.g., the company internet going down, a visit by a boss, a joint company meeting, a fire drill, a visit to a customer, et cetera) do have the potential to completely reset the interaction dynamics for the day. This is both interesting from an empirical point of view and important for the development of theory of how human interaction is shaped, maintained, and developed.

## Supporting information

**S1 File. Results for the numerical evaluation of the moving window REM with fixed and data-driven window widths.**
(PDF)

## Author Contributions

**Conceptualization:** Marlyne Meijerink-Bosman, Roger Leenders, Joris Mulder.

**Funding acquisition:** Joris Mulder.

**Investigation:** Marlyne Meijerink-Bosman.

**Methodology:** Marlyne Meijerink-Bosman, Roger Leenders, Joris Mulder.

**Visualization:** Marlyne Meijerink-Bosman.

**Writing – original draft:** Marlyne Meijerink-Bosman.

**Writing – review & editing:** Roger Leenders, Joris Mulder.

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
