## [Decision Letter · Decision Letter 0]

23 May 2022

PONE-D-22-05391Dynamic relational event modeling: Testing, exploring, and

applyingPLOS ONE

Dear Dr. Marlyne,

Thank you for submitting your manuscript to PLOS ONE. After careful consideration, we feel that it has merit but does not fully meet PLOS ONE’s publication criteria as it currently stands. Therefore, we invite you to submit a revised version of the manuscript that addresses the points raised during the review process.

We look forward to receiving your revised manuscript.

Kind regards,

Lei Shi

Academic Editor

PLOS ONE

Journal Requirements:

Reviewers' comments:

Reviewer's Responses to Questions

**Comments to the Author**

1. Is the manuscript technically sound, and do the data support the conclusions?

Reviewer #1: Yes

Reviewer #2: Yes

2. Has the statistical analysis been performed appropriately and rigorously? 

Reviewer #1: Yes

Reviewer #2: Yes

3. Have the authors made all data underlying the findings in their manuscript fully available?

Reviewer #1: Yes

Reviewer #2: Yes

4. Is the manuscript presented in an intelligible fashion and written in standard English?

Reviewer #1: Yes

Reviewer #2: Yes

5. Review Comments to the Author

Reviewer #1: The paper is a lucid and well-executed methodological article, providing a well-validated method to study temporal parameter heterogeneity in relational event models. It should be of interest to a wide-range of social scientists interested in studying network dynamics. Congratulations on a solid piece of work.

Reviewer #2: This paper extends the REM framework with approaches for testing and exploring time-varying network effects and uses the Bayesian approach and moving window of sliding relationship event history to study time-varying network effects. In addition, the method of this study is applied to an empirical application. This paper finds that estimation with empirically determined window widths achieves both good accuracy for time intervals with important changes and good precision for time intervals with hardly any changes in the effects.

The model in this paper is logical and the results are abundant, interesting and convincing. The conclusions have strong practical significance and reference value. It's a great honor to read this paper, but there are still a few suggestions before publication:

1. In the "Recency", should factors other than email be considered? Such as making calls, sending messages, etc.

2. In addition, whether employees' face-to-face interactions are affected by other factors, such as online social relationships.

3. There is a mistake in the use of punctuation in the text, e.g., the sentence on page 25, “have been introduced [e.g.,][]Butts2008,Nooy2011,Perry2013,Stadtfeld2017,Vu2011. Generally, ……”, here I think “have been introduced (e.g., [1,8,22]) Generally, ……” may be correct. Furthermore, Nooy2011 and Stadtfeld2017 were not found in ref.

6. PLOS authors have the option to publish the peer review history of their article (what does this mean?). If published, this will include your full peer review and any attached files.

Reviewer #1: No

Reviewer #2: No

---

## [Author Response · Author response to Decision Letter 0]

16 Jun 2022

Response to reviewer 1: Thank you for the positive comments on our manuscript. 

Response to reviewer 2: Thank you for the positive comments on our manuscript and suggestions on how we could improve our work. 

In response to comment 1 and 2: We agree with you that adding information about additional types of interactions between the employees has the potential to lead to a more rich and detailed understanding of the dynamics of the face-to-face contacts. Unfortunately, we are restricted due to the limited available information in the data. Therefore, we cannot extend the analysis as suggested. We acknowledge this in the Discussion section of the revised manuscript, and state that the proposed model could be extended if such information would be available:

"Moreover, in the current analysis we focus on modeling the predictors of face-to-face interactions between employees in the office space. In many organizations, employees interact via different modes of communication, e.g., face-to-face interactions, e-mail messages, phone calls, etc. In addition, employees may develop social relationships outside of the office space. All these factors are likely to affect the patterns of observed face-to-face interactions between employees in the office space. In the current analysis, we are restricted due to the limited available information in the data. When such information would be available, however, it is recommended to consider it in the configuration of the predictor variables to obtain a more detailed understanding of the dynamics of employees’ face-to-face interactions (e.g., see [27])."

In response to comment 3: Thank you for making us aware of these mistakes. We have corrected them in the revised manuscript.

We hope all your comments have been satisfactorily addressed. Thank you again for your review.

---

## [Decision Letter · Decision Letter 1]

18 Jul 2022

Dynamic relational event modeling: Testing, exploring, and applying

PONE-D-22-05391R1

Dear Dr. Marlyne Meijerink-Bosman,

We’re pleased to inform you that your manuscript has been judged scientifically suitable for publication and will be formally accepted for publication once it meets all outstanding technical requirements.

Kind regards,

Lei Shi

Academic Editor

PLOS ONE

Additional Editor Comments (optional):

Reviewers' comments:

Reviewer's Responses to Questions

**Comments to the Author**

1. If the authors have adequately addressed your comments raised in a previous round of review and you feel that this manuscript is now acceptable for publication, you may indicate that here to bypass the “Comments to the Author” section, enter your conflict of interest statement in the “Confidential to Editor” section, and submit your "Accept" recommendation.

Reviewer #2: All comments have been addressed

2. Is the manuscript technically sound, and do the data support the conclusions?

Reviewer #2: Yes

3. Has the statistical analysis been performed appropriately and rigorously? 

Reviewer #2: Yes

4. Have the authors made all data underlying the findings in their manuscript fully available?

Reviewer #2: Yes

5. Is the manuscript presented in an intelligible fashion and written in standard English?

Reviewer #2: Yes

6. Review Comments to the Author

Reviewer #2: The authors of this paper have given a very reasonable explanation and perfect solution to the modification opinions. I am pleased to recommend its publication.

7. PLOS authors have the option to publish the peer review history of their article (what does this mean?). If published, this will include your full peer review and any attached files.

Reviewer #2: No

---

## [Editor Report · Acceptance letter]

22 Jul 2022

PONE-D-22-05391R1 

Dynamic relational event modeling: Testing, exploring, and applying 

Dear Dr. Meijerink-Bosman:

I'm pleased to inform you that your manuscript has been deemed suitable for publication in PLOS ONE. Congratulations! Your manuscript is now with our production department. 

Kind regards, 

on behalf of

Dr. Lei Shi 

Academic Editor

PLOS ONE